# Advancing 3D Object Grounding Beyond a Single 3D Scene

## ABSTRACT

As a widely explored multi-modal task, 3D object grounding endeavors to localize a unique pre-existing object within a single 3D scene given a natural language description. However, such a strict setting is unnatural as it is not always possible to know whether a target object actually exists in a specific 3D scene. In real-world scenarios, a collection of 3D scenes is generally available, some of which may not contain the described object while some potentially contain multiple target objects. To this end, we introduce a more realistic setting, named Group-wise 3D Object Grounding, to simultaneously process a group of related 3D scenes, allowing a flexible number of target objects to exist in each scene. Instead of localizing target objects in each scene individually, we argue that ignoring the rich visual information contained in other related 3D scenes within the same group may lead to sub-optimal results. To achieve more accurate localization, we propose a baseline method named GNL3D, a Grouped Neural Listener for 3D grounding in the group-wise setting, which extends the traditional 3D object grounding pipeline with a novel language-guided consensus aggregation and distribution mechanism to explicitly exploit the intra-group visual connections. Specifically, based on context-aware spatial-semantic alignment, a Language-guided Consensus Aggregation Module (LCAM) is developed to aggregate the visual features of target objects in each 3D scene to form a visual consensus representation, which is then distributed and injected into a consensus-modulated feature refinement module for refining visual features, thus benefiting the subsequent multi-modal reasoning. Furthermore, we design a curriculum strategy to promote the LCAM to learn step by step how to extract effective visual consensus with the existence of negative 3D scenes where no target object exists. To validate the effectiveness of the proposed method, we reorganize and enhance the ReferIt3D dataset and propose evaluation metrics to benchmark prior work and GNL3D. Extensive experiments demonstrate GNL3D achieves state-of-the-art results on both the group-wise setting and the traditional 3D object grounding task.

## CCS CONCEPTS

• **Computing methodologies** → **Artificial intelligence**; **Computer vision**; **Computer vision tasks**.

## KEYWORDS

3D object grounding, group-wise learning, curriculum learning

**Query:** Find the monitor that is on top of a desk.

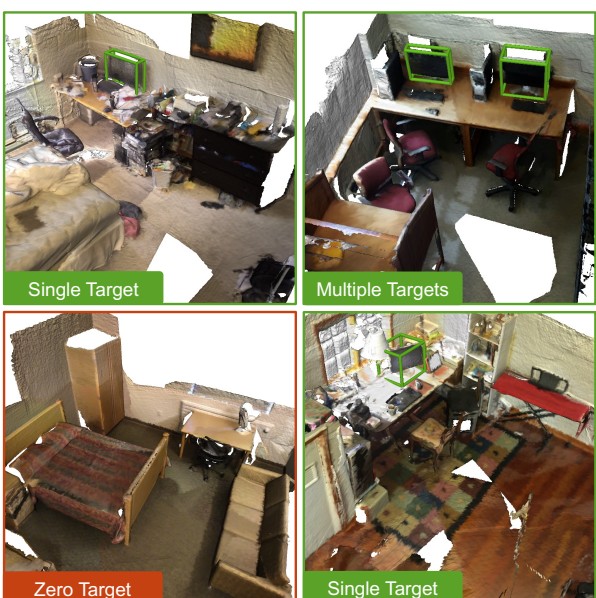

**Figure 1: Group-wise 3D Object Grounding, where a flexible number of target objects (zero, single or multiple) should be localized in a group of 3D scenes for a given description.**

**ACM Reference Format:**
Anonymous Author(s). 2024. Advancing 3D Object Grounding Beyond a Single 3D Scene. In *Proceedings of the 32nd ACM International Conference on Multimedia (MM'24), October 28-November 1, 2024, Melbourne, Australia.* ACM, New York, NY, USA, 10 pages. https://doi.org/10.1145/nnnnnnn.nnnnnnn

## 1 INTRODUCTION

Grounding target objects described by users in 3D environments is a fundamental capability that facilitates various multimedia applications, such as intelligent robot service, smart home, and metaverse. Following this demand, 3D object grounding [2, 6] has recently become a research hotspot, attracting wide attention from researchers in various fields [5, 10, 18, 21, 26, 46, 49, 61, 63]. Despite the significant advancements made in this field, the existing 3D object grounding task is overly idealistic, which aims to localize a unique target object that has been known to exist in a single 3D scene. This setting limits the applicability of 3D object grounding in real-world scenarios. In general, a group of related 3D scenes is available (e.g., a house comprises several rooms), some of which may not contain the described object while some may potentially contain multiple target objects (as illustrated in Fig. 1).

To address this limitation, in this paper, we introduce a new realistic setting, namely Group-wise 3D Object Grounding, where a flexible number of target objects should be localized in a group of

related 3D scenes given a referring sentence. The proposed group-wise 3D object grounding setting generalizes the traditional 3D object grounding task in the following two aspects: (1) Compared with the traditional single target assumption, our setting allows for the existence of zero or multiple target objects in a specific 3D scene. (2) The proposed setting requires querying a group of related 3D scenes simultaneously with a single referring expression. Note that the application scope of group-wise 3D object grounding is the superset of the traditional task, which means (1) or (2) can be omitted based on the actual situations, and thus our setting is more natural and practical in broader real-world scenarios.

Recently, Multi3DRefer [59] has proposed an effective approach to localize a flexible number of objects in a single 3D scene given a text description. It formulates the problem as a multi-label task and adapts existing 3D single object grounding methods by replacing the loss function with a binary cross-entropy loss. Based on this method, a straightforward solution to further address group-wise 3D object grounding is to simply apply the adapted models to each individual 3D scene in a group and integrate these results for group-wise grounding. However, such a direct solution would result in inferior performance, as it localizes target objects in each scene individually based solely on the linguistic expression, failing to capture the rich visual information contained in other related 3D scenes within the same group. In comparison to the linguistic description, the visual clues of target objects can bridge the modality gap and provide a more precise concept about the described object's properties and its spatial relations to anchor objects. Ignoring such intra-group visual connections may lead to sub-optimal results. Nevertheless, it is challenging to capture effective visual clues of target objects from a group of 3D scenes. Firstly, in addition to the common properties described by the same referring sentence, target objects in each scene also have their own unique attributes and diverse context objects. Without proper architecture design, these unique visual characteristics that are not mentioned in text can mislead the grounding results of other scenes and limit the overall performance. Moreover, there might be negative 3D scenes in a group where no target object exists. These negative scenes cannot provide direct visual clues about the described object, which makes it even more challenging to extract valid intra-group visual information to form an accurate target concept.

To tackle the aforementioned issues, we propose a baseline method named GNL3D, a Grouped Neural Listener for 3D grounding in the group-wise setting, which facilitates a simultaneous processing of multiple input 3D scenes with a language description and generates localization results for all target objects. GNL3D adapts the typical 3D object grounding pipeline [63] with a Language-guided Consensus Aggregation and Distribution (LCAD) mechanism to explicitly leverage language and intra-group visual information. In our proposed Language-guided Consensus Aggregation Module (LCAM), the visual features of target objects in each 3D scene are adaptively aggregated to form a visual consensus representation under the guidance of the language description. Specifically, we employ the language-objects cross-attention mechanism to select language-relevant visual information and consider both the semantic-level and spatial contextual visual features to assemble a comprehensive representation of the described object's common properties, thus avoiding the ambiguity and incompleteness caused

by irrelevant or partial information. In addition, we apply a curriculum learning strategy to train our model, which gradually shifts the input contextual visual features from the subgroup that only contains positive scenes to the entire group, thus promoting the LCAM to learn how to extract effective visual consensus with the existence of negative 3D scenes step by step. After the visual consensus extraction, we further devise a Consensus-modulated Feature Refinement Module (CFRM), which can dynamically adjust the behavior of a spatial transformer encoder by distributing and injecting the homo-modal visual consensus information to produce consensus-relevant visual features. In the proposed CFRM, we first situate the consensus features within the specific context of each 3D scene and then develop a consensus-aware dynamic spatial attention mechanism to refine the visual features, which benefits the subsequent multi-modal alignment between the text and the relevant region to make a correct prediction. Besides, we design two types of prediction heads for GNL3D to perform 3D grounding at both scene-level and object-level to better adapt to the group-wise setting. To facilitate a systematic evaluation, we reorganize and enhance the ReferIt3D [2] dataset and propose evaluation metrics to benchmark prior work and GNL3D.

Our main contributions can be summarized as follows:

- We formalize a novel Group-wise 3D Object Grounding setting to explore the flexible number object grounding in a group of 3D scenes, which advances user-specified 3D object grounding towards more practical applications.
- We propose GNL3D as a strong research baseline for the challenging setting. It uses a new language-guided consensus aggregation and distribution mechanism to explicitly exploit language and intra-group visual connections for better performance.
- We develop a language-guided consensus aggregation module with curriculum learning to effectively extract visual consensus features and overcome the challenges caused by irrelevant information and negative scenes. A consensus-modulated feature refinement module is also devised to fully leverage the homo-modal visual consensus for refining visual features.
- Extensive experiments validate the effectiveness of our proposed baseline method GNL3D, which achieves state-of-the-art results not only on the group-wise setting, but also on the traditional 3D object grounding task.

## 2 RELATED WORK

### 2.1 3D Object Grounding

Grounding natural language in 3D environments is a fundamental task in the vision-language understanding field, which may enable wide-ranging multimedia applications, such as embodied intelligence [13], smart home [62], and AR/VR/metaverse [12, 37]. As the pioneer works, Scanrefer [6] and Referit3D [2] propose two datasets consisting of language descriptions of 3D objects from the real-world dataset ScanNet [11] and introduce the 3D object grounding task, aiming to localize a unique target object in a single 3D scene using language. In detail, the ReferIt3D [2] dataset contains both template-based descriptions generated based on spatial relations between objects (Sr3D) and human-annotated fine-grained descriptions (Nr3D). Different approaches have been proposed to tackle this task [1, 4, 19, 23, 32, 33, 36, 46, 49, 52]. The prevalent

methods solve 3D object grounding with a two-stage pipeline: they leverage off-the-shelf 3D object detectors [34, 38] or segmentors [9, 24, 44] to obtain the object proposals, utilize text and point cloud encoders [39, 40] to extract features, and then employ various cross-modal fusion or matching mechanisms to select the target object. Early works [2, 16, 20, 53] use graph representations to model the relations among objects. With the development of transformers [43], recent works [3, 5, 7, 10, 17, 18, 21, 26, 41, 50, 61, 63] have adopted transformers for multi-modal feature fusion. Among them, 3DVG-Transformer [61] and ViL3DRel [10] focus on improving 3D spatial relation modeling. SAT [50], MVT [21], LAR [3] and ViewRefer [17] leverage multi-view images and 2D semantics for better performance. 3DJCG [5], D3Net [8], UniT3D [7], and 3D-VisTA [63] are unified frameworks that can address both 3D grounding and captioning. Recently, [22] proposes a method named 3DOGS-Former to explore paragraph-level dense object grounding in a 3D scene. [59] introduces a new task called Multi3DRefer, which generalizes the traditional 3D object grounding to a flexible number of target objects given sentence descriptions. In contrast to prior works that only localize target objects in a single 3D scene, our approach focuses on querying a group of related 3D scenes simultaneously, advancing user-specified 3D object grounding towards more practical applications.

## 2.2 Group-wise 2D Object Localization

Localizing objects of interest within a group of 2D images has been widely studied for a long time. Early works focus on image co-localization [27, 31, 42, 45], which aims to localize objects of the same class emerging in a set of distinct images with bounding boxes. Co-Salient Object Detection (Co-SOD) is a more recent research focus [14, 15, 25, 29, 47, 48, 51, 54–58, 60, 64], where the common salient objects across a group of relevant images should be detected with segmentation maps. In Co-SOD, the target object does not need to be specified by a language description, but it is required to appear commonly in all images. Many impressive Co-SOD approaches follow a unified aggregation-and-distribution paradigm, which first aggregate all image features in the group to form a consensus representation and then distributing it back to each image feature. For instance, [60] sums up all features for aggregation and utilizes a gradient feedback mechanism for distribution. [15] generates a consensus attention map with an group affinity module and multiplies it back to the individual image features. [25] uses a pairwise similarity map to form the intra-group consensus information and conducts the distribution via a dense correlation module. [51] first obtains consensus seeds via affinity maps and then propagated the seeds using normalized convolution operations. [57] applies convolution and self-attention operations to encode the consensus cues into a series of kernels and implements the distribution process by dynamic convolution. [48] utilizes transformer-based modules to perform aggregation at the semantic level, and proposes image-specific distribution while maintaining a consistent consensus representation. However, existing aggregation-and-distribution methods designed for Co-SOD cannot be directly applied to the group-wise 3D object grounding task. Firstly, unlike the semantic-level consensus aggregation using pure visual modality in Co-SOD, target objects in a group of 3D scenes are described by natural

language and possess complex contextual relationships with other objects in the same scene. Secondly, while Co-SOD requires every image in a group to contain the target object, there can be negative 3D scenes in group-wise 3D object grounding, making it even harder to explore the intra-group visual connections in our setting.

## 3 METHOD

### 3.1 Problem Formulation

Given a group of related 3D scenes and an utterance that describes an object, the goal of group-wise 3D object grounding is to simultaneously localize all corresponding target objects in each scene. Following the setup of ReferIt3D [2], we assume access to a set of $M_k$ objects for each 3D scene $\mathcal{S}_k = \{O_{k1}, \cdots, O_{kM_k}\}$ in the group ($k = 1, \cdots, K$), where each object is represented as a point cloud $O \in \mathbb{R}^{N \times (3+C)}$ of $N$ points with their 3D coordinates and $C$-dim auxiliary feature such as color and normal vectors. We denote the word-tokenized utterance as $\mathcal{U} = \{w_i\}_{i=1}^{T}$, where $w_i$ represents the $i$-th word and $T$ is the total number of words in the sentence. The expected output for the $k$-th 3D scene is the set of target referred objects $\mathcal{T}_k \subset \mathcal{S}_k$. In particular, the number of target objects in each scene is flexible, $i.e.$, there can be not only positive 3D scenes containing one or multiple objects that match the description but also negative ones where no referred object exists.

### 3.2 Overview

In this section, we introduce the overall architecture of our proposed GNL3D, a Grouped Neural Listener for 3D grounding in the group-wise setting. As illustrated in Fig. 2, there are five main components in GNL3D, including text encoding, scene encoding, a Language-guided Consensus Aggregation Module (LCAM), a Consensus-modulated Feature Refinement Module (CFRM) and a multi-modal fusion decoder.

**Scene-Text Encoding.** We follow 3D-VisTA [63] to perform 3D scene and text encoding. For each 3D scene, a collection of object point clouds $\mathcal{S}_k = \{O_{k1}, \cdots, O_{kM_k}\}$ is given. For each 3D object point cloud $O_{ki} \in \mathcal{S}_k$, the scene embedding layer first extracts the semantic-level object feature vector $F_{ki}^{S} \in \mathbb{R}^D$ and its semantic class through a PointNet++ backbone [40], where $D$ is the dimensionality of the feature. Then, we feed the semantic-level object features $F_k^S = \{F_{k1}^S, \cdots, F_{kM_k}^S\}$ into a spatial transformer encoder [10, 63] to enhance the contextual relationships between objects in each 3D scene and obtain the spatial contextual object features $F_k^C \in \mathbb{R}^{M_k \times D}$. As for the text encoding, we encode the input textual utterance $\mathcal{U}$ with $T$ word tokens via a four-layer transformer encoder, which is initialized by the first four layers of a pre-trained BERT [28]. The resulting linguistic features can be represented as $L = \{l_{\text{cls}}, l_1, \cdots, l_T\} \in \mathbb{R}^{(T+1) \times D}$, where $l_{\text{cls}}$ is the output of a special classification token ([CLS]).

**LCAM & CFRM.** Our model utilizes a novel Language-guided Consensus Aggregation and Distribution (LCAD) mechanism to explicitly exploit the intra-group visual connections. Specifically, the proposed LCAD mechanism comprises two steps. Firstly, under the guidance of the linguistic features, the encoded semantic-level and spatial contextual object features in all 3D scenes are fed into LCAM (Sec. 3.3) to extract visual consensus features, which capture the

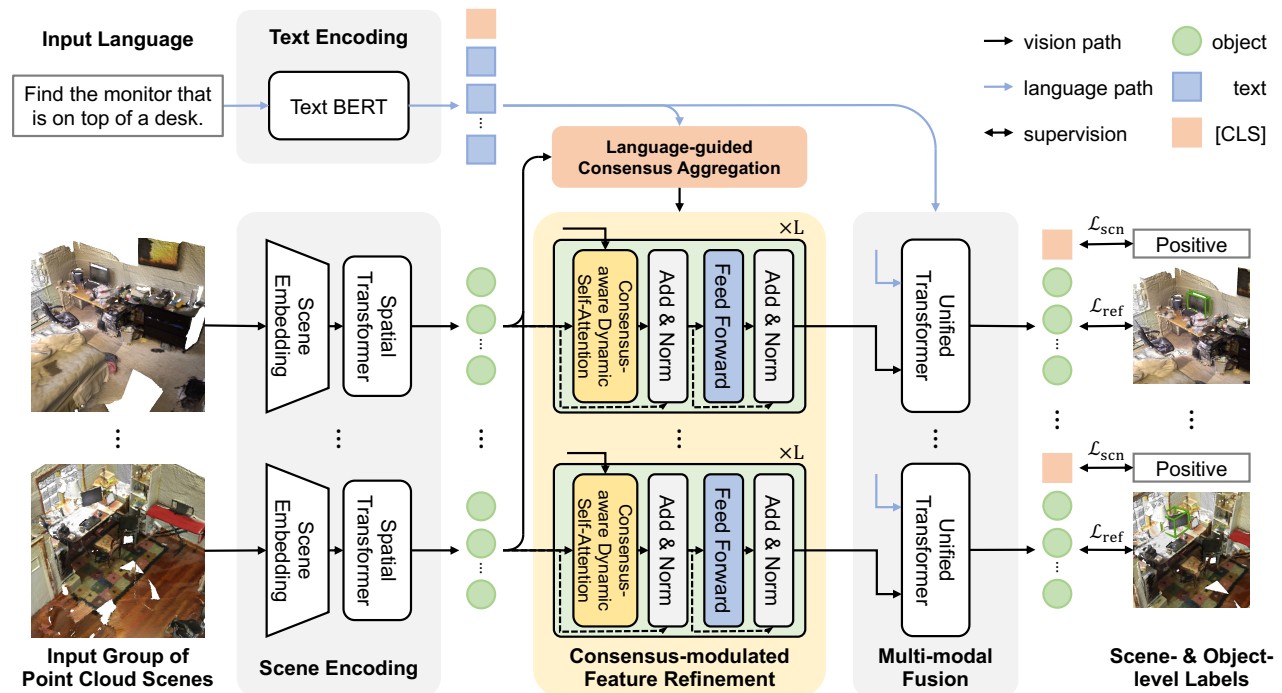

**Figure 2: The overall architecture of our GNL3D framework, which includes scene-text encoding, language-guided consensus aggregation, consensus-modulated feature refinement and multi-modal fusion modules. With the help of the language-guided consensus aggregation and distribution mechanism, the intra-group visual connections can be explicitly exploited.**

described object's common visual and relational properties to form a precise target concept. Subsequently, these visual consensus features are distributed back to the spatial contextual object features in each 3D scene via CFRM (Sec. 3.4) for discriminability enhancement, which benefits the subsequent multi-modal reasoning.

**Multi-modal Fusion.** For each 3D scene, we simply concatenate its refined contextual object features $F_k^R \in \mathbb{R}^{M_k \times D}$ with the linguistic features $L$ to form a joint sequence $\{l_{\text{cls}}, l_1, \cdots, l_T, F_{k1}^R, \cdots, F_{kM_k}^R\}$. We add learnable type embeddings to these tokens to differentiate text and visual modalities. Next, a multi-layer unified transformer [43, 63] is applied to perform multi-modal fusion and reasoning on the joint sequence. Finally, we develop two types of prediction heads to perform 3D grounding at both scene- and object-level. More details about decoding will be introduced in Sec. 3.5.

### 3.3 Language-guided Consensus Aggregation

Due to the inherent modality gap, directly localizing target objects in each 3D scene individually based solely on the linguistic features often results in inferior performance. We resort to intra-group homo-modal visual consensus features to bridge the modality gap and provide a more precise concept about the described object's common visual and relational properties. Different from the semantic-level consensus aggregation using pure visual modality in the traditional 2D group-wise learning, it is more challenging to extract effective intra-group visual clues in the proposed group-wise 3D object grounding task: (1) the visual consensus features

should capture comprehensive common characteristics of target objects that are mentioned in natural language in all positive 3D scenes, which requires fine-grained cross-modality alignment and interaction considering the complex spatial contextual relationships between objects in each 3D scene; (2) there can be negative 3D scenes in a group that are not known in advance and cannot provide direct visual information about the described object, which require to be handled properly in the aggregation process. To address the above challenges, we first devise the LCAM to adaptively aggregate visual consensus features from both semantic and contextual perspectives under the guidance of the linguistic features based on a dual-stream transformer decoder. Then, we develop a Curriculum Learning (CL) strategy to facilitate the training process of the LCAM in the presence of negative 3D scenes step by step, as illustrated in Fig. 3.

**Semantic-level Consensus Aggregation.** We directly concatenate the encoded semantic-level object features in all 3D scenes and take the resulting sequence $F^S = \{F_1^S, \cdots, F_K^S\} \in \mathbb{R}^{(M_1 + \cdots + M_K) \times D}$ together with the linguistic features $L \in \mathbb{R}^{(T+1) \times D}$ as input to the semantic branch of LCAM. In practice, each branch of LCAM is based on a $N_L$-layer standard Transformer decoder [43], each layer of which includes self-attention and cross-attention blocks followed by a feed-forward neural network. In LCAM, the linguistic features serve as initial queries, which adaptively aggregate relevant visual information through language-objects cross-attention blocks and gradually update themselves layer by layer to produce the final visual consensus features. At this branch, the final representation

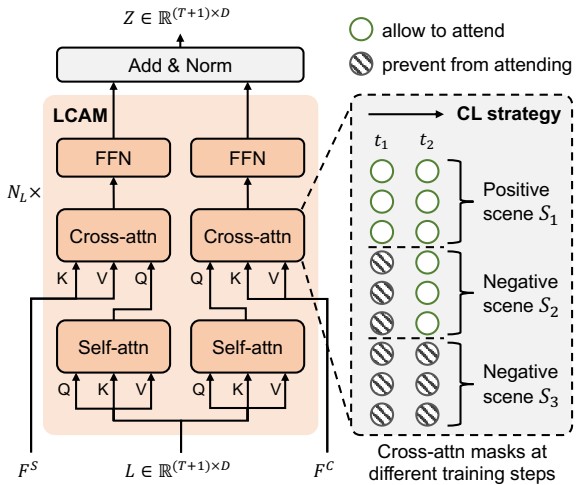

Figure 3: Illustration of LCAM that aggregates visual consensus features from both semantic and contextual perspectives. The CL strategy facilitates its training process progressively.

of each word token aggregates visual features of the corresponding category of objects in all 3D scenes. The output features can be denoted as semantic-level visual consensus features $Z^S \in \mathbb{R}^{(T+1) \times D}$, which only capture non-contextual intra-group visual information, since the input object features are from the scene embedding layer before attending to surrounding objects.

**Contextual Consensus Aggregation.** To capture the described object's common relational visual properties, we further feed the concatenation of all contextual object features $F^C = \{F_1^C, \cdots, F_K^C\}$ into the contextual branch of LCAM and utilize linguistic queries $L$ to adaptively aggregate relevant information from the sequence. The process is similar to the semantic-level aggregation, however, while it is very likely that every 3D scene in the group contains objects of the same category as the object being described, not all scenes are positive scenes that can provide valid contextual visual clues of target objects. To facilitate the training process of LCAM, we develop a CL strategy based on a mask mechanism for the contextual branch. Specifically, for each layer of the transformer decoder, the multi-head cross-attention takes the hidden representations $H$ from the previous self-attention block as the query $Q = HW_q$, and the concatenated contextual object features $F^C$ are used as both the key $K = F^C W_k$ and the value $V = F^C W_v$. The output of a cross-attention head $A$ is computed as:

$$A = \text{softmax}(QK^\top / \sqrt{d} + M(t))V \tag{1}$$

$$M(t)_{ij} = \begin{cases} 0, & \text{allow to attend} \\ -\infty, & \text{prevent from attending} \end{cases} \tag{2}$$

where $d$ is the dimension of the embedding space, $W_q, W_k, W_v \in \mathbb{R}^{D \times d}$ are projection matrices, and $M(t) \in \mathbb{R}^{(T+1) \times (M_1 + \cdots + M_K)}$ is the mask matrix at the training step $t$ that determines whether a pair of tokens can be attended to each other.

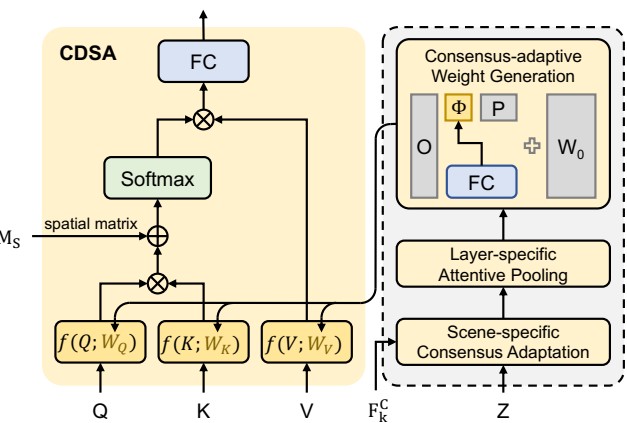

Figure 4: The detailed architecture of CDSA. The left part shows the adapted attention block in the CFRM, and the right part shows the process of dynamic weight generation.

**Curriculum Learning Strategies.** In the initial training stage, only the elements corresponding to positive 3D scenes in the mask matrix are 0s, indicating that the contextual branch only focuses on the critical part of the input group at this stage. With the training step $t$ increasing, what the queries can attend to gradually shifts from the part only containing positive scenes to the entire sequence. Concretely, we divide the training process into $\kappa$ stages, and define three different pacing functions (linear, exponential or logarithmic) to control how the training step $t \in (0, \kappa)$ increases. At the training step $t$, $t/\kappa$ negative scenes in the group can be attended to, and newly added negative scenes are randomly selected.

After obtaining the contextual visual consensus features $Z^C$, the output features of the two branches of LCAM are added together to get the final visual consensus features $Z = Z^S + Z^C \in \mathbb{R}^{(T+1) \times D}$, which assemble a comprehensive representation of the intra-group visual connections.

### 3.4 Consensus-modulated Feature Refinement

In order to fully exploit the extracted visual consensus features, we distribute them back to the contextual object features in each 3D scene for feature enhancement. There are many methods for consensus distribution, such as direct concatenation, linear addition, cross-attention, etc. However, we empirically find that the dynamic attention-based consensus distribution is more effective. Our CFRM extends a spatial transformer encoder [63] by replacing the traditional self-attention with a **Consensus-aware Dynamic Spatial Attention (CDSA)** mechanism, as shown in Fig. 4. Specifically, we generate consensus-adaptive weights to produce the query $Q$, key $K$, and value $V$ in CDSA conditioned on the visual consensus features $Z$, which can be represented as:

$$Q = f(X; W_Q), K = f(X; W_K), V = f(X; W_V), \tag{3}$$

where $f(\cdot; W)$ indicates linear projection parameterized by $W$, $X$ represents the input visual features, and $W_Q, W_K, W_V \in \mathbb{R}^{D_{in} \times D_{out}}$ are the dynamic projection weights. Besides, we guide the attention calculate in CDSA by adding a spatial proximity matrix [61]

to enhance the 3D spatial relation understanding. The process of generating the dynamic weights comprises three steps, *i.e.*, the scene-specific consensus adaptation, the layer-specific attentive pooling, and the consensus-adaptive weight generation.

**Scene-specific Consensus Adaptation.** To better cope with the variation of context objects in different 3D scenes, we first situate the visual consensus features $Z$ within the specific context of each 3D scene via a plain multi-head cross-attention layer [43], as $Z_k =$ CrossAttn$(Z, F_k^C, F_k^C) + Z$, where $F_k^C \in \mathbb{R}^{M_k \times D}$ is the contextual object features and $Z_k \in \mathbb{R}^{(T+1) \times D}$ is the output scene-specific visual consensus features for the $k$-th scene.

**Layer-specific Attentive Pooling.** Considering the visual consensus features correspond to a different number of tokens and each layer of the CFRM may prefer different tokens, we subsequently introduce a learnable layer-specific embedding $e_i \in \mathbb{R}^D$ for each layer $i$ of the CFRM to extract layer-specific visual consensus features $Z_{k,i} \in \mathbb{R}^D$ dynamically, which can improve the model flexibility at negligible cost. In practice, the attentive pooling is based on a multi-head cross-attention layer [43], as $Z_{k,i} = $ CrossAttn$(e_i, Z_k, Z_k)$.

**Consensus-adaptive Weight Generation.** Considering that the dynamic weights are in a high-dimensional space of $D_{\text{in}} \times D_{\text{out}}$, directly generating them using fully-connected layers is unaffordable. Motivated by the dynamic channel fusion [30], we generate dynamic weights following the matrix decomposition paradigm. Taking the $i$-th layer of CFRM for the $k$-th 3D scene as an example, the proposed consensus-adaptive weight generation can be formulated as:

$$[W_Q^i, W_K^i, W_V^i] = W_0^i + O\Phi(Z_{k,i})P^{\mathsf{T}}, \qquad (4)$$

where $W_0^i \in \mathbb{R}^{D_{\text{in}} \times D_{\text{out}}}$ is the layer-specific static learnable weights. $O \in \mathbb{R}^{D_{\text{in}} \times d_{\text{w}}}$ and $P \in \mathbb{R}^{d_{\text{w}} \times D_{\text{out}}}$ are also static learnable weights, but sharable across all CFRM layers to reduce the parameter numbers and prevent the model from overfitting. $\Phi(Z_{k,i})$ is a fully-connected layer, which produces a dynamic matrix of shape $d_{\text{w}} \times d_{\text{w}}$ with the dimension reduction ratio $r = D/d_{\text{w}}$.

## 3.5 Training and Inference

Given the output of the multi-modal fusion module for each 3D scene, which can be denoted as $\{f_{\text{cls}}^w, f_{1:T}^w, f_{1:M_k}^o\}$ for [CLS], text tokens, and 3D object tokens, respectively, we develop two types of prediction heads based on two-layer MLPs to perform 3D grounding at both scene- and object-level.

**Scene-level Grounding.** The scene-level grounding head takes the $f_{\text{cls}}^w$ as the global representation to generate a scalar score and applies a sigmoid function to obtain a probability $p_k^s$ indicating whether the $k$-th 3D scene contains the described object. We supervise this head with a binary cross-entropy loss $\mathcal{L}_{\text{scn}}$.

**Object-level Grounding.** Similarly, given the output embeddings of the object tokens $f_{1:M_k}^o$, the object-level grounding head generates a scalar score for each object proposal $O_{ki}$. In the original single-target setting where only one target object can exist in a positive scene, a softmax function is adopted to calculate the probability $p_{ki}^o$ and a cross-entropy loss $\mathcal{L}_{\text{ref}}$ is used for training. In the multi-target setting, we use a sigmoid function to obtain the score and adopt a binary cross-entropy loss for $\mathcal{L}_{\text{ref}}$.

**Inference.** At inference time, we first take all 3D scenes with the predicted scene-level scores $p_k^s$ above a threshold $\tau_s$ as positive scenes. We then make object-level predictions for each positive 3D scene. In the single-target setting, we take the object proposal with the maximum probability as the target object. In the multi-target setting, all object proposals with predicted scores $p_{ki}^o$ above a threshold $\tau_o$ are predicted as target objects.

## 4 EXPERIMENTS

### 4.1 Datasets

**Sr3D.** We evaluate GNL3D and compare it to prior work on the ReferIt3D benchmark [2] for the traditional 3D object grounding task. We specifically focus on the Sr3D dataset, which is constructed using "target-relation-anchor" templates to automatically generate sentences. The sentences utilize spatial relations to distinguish objects of the same class. Sr3D is split into "easy" and "hard" subsets in evaluation. The "view-dep." and "view-indep." subsets depend on whether the description is related to the speaker's view.

**G-Sr3D-ST.** Given that there exists some repeated "target-relation-anchor" sentences in different 3D scenes, we reconstruct Sr3D to the form of "one sentence vs. a group of referred 3D scenes", named as G-Sr3D-ST, where each positive scene in a group only contains a Single Target object. For each group, we randomly sample negative 3D scenes from other groups, and verify the validity by ensuring that the semantic class of the target object or anchor objects for the description do not appear in these negative scenes. The re-built dataset has 65200 sentence-scene group pairs, and each scene group includes up to 8 positive scenes. We also include some groups containing only one scene. For multi-scene groups, the positive to negative sample ratio is 1 : 1 in evaluation.

**G-Sr3D-MT.** To comprehensively evaluate GNL3D's performance, we also introduce the G-Sr3D-MT dataset as a supplement, where each positive scene can contain Multiple Target objects. We propose two approaches to construct G-Sr3D-MT. Firstly, we extend G-Sr3D-ST to the multi-target setting by randomly concatenating multiple (up to 4) positive 3D scenes in each sentence-scene group pair and removing unnecessary objects to form a multi-target scene of appropriate size. Secondly, we enhance Sr3D by synthesizing additional multi-target utterances using the following compositional template: "number-target-relation-anchor", e.g., "find two chairs closest to the door". Negative scenes are collected in the same way as G-Sr3D-ST. Detailed statistics can be found in Tab. 2.

### 4.2 Experimental Settings

**Evaluation Metrics.** We evaluate models for these three datasets under different evaluation metrics. In the default setting of Sr3D, ground-truth object proposals are provided and there is only one target object in a single 3D scene. The metric is the accuracy of selecting the target bounding box among the proposals. For the G-Sr3D-ST setting, we adopt the metric of mean accuracy (mAcc) of a 3D scene group. As for the G-Sr3D-MT setting where each scene contains a flexible number of target objects, we measure the F1 score for each scene in a group following the previous work [59] and report the average score as the mean F1 (mF1) metric. To investigate model performance for different group-wise scenarios, we consider the following 4 cases: a) single target (ST) w/o negative

**Table 1: Comparison results on the group-wise setting. We report mAcc and mF1 for G-Sr3D-ST and -MT respectively.**

| Method | Type | ST w/o Negatives | | | ST w/ Negatives | | | MT w/o Negatives | | | MT w/ Negatives | | |
|---|---|---|---|---|---|---|---|---|---|---|---|---|---|
| | | Single | Multiple | **All** | Single | Multiple | **All** | Single | Multiple | **All** | Single | Multiple | **All** |
| 3DVG-Trans [61] | Individual | 48.2 | 53.3 | 51.1 | 51.0 | 55.2 | 53.4 | 48.8 | 49.0 | 48.9 | 44.3 | 47.5 | 46.3 |
| TransRefer3D [18] | | 52.9 | 60.0 | 56.9 | 55.4 | 61.0 | 58.5 | 55.4 | 53.7 | 54.3 | 52.7 | 50.6 | 51.4 |
| MVT [21] | | 58.4 | 67.9 | 63.9 | 62.9 | 68.7 | 66.2 | 60.8 | 60.3 | 60.5 | 57.1 | 55.6 | 56.1 |
| 3D-VisTA [63] | | 65.7 | 72.6 | 69.7 | 67.9 | 73.8 | 71.0 | 63.9 | 65.6 | 64.9 | 59.7 | 59.4 | 59.5 |
| 3D-VisTA + GSA | Grouped | 65.6 | 70.5 | 68.4 | 68.4 | 72.9 | 70.9 | 63.4 | 64.1 | 63.8 | 58.5 | 58.0 | 58.2 |
| GNL3D (Ours) | | **69.9** | **78.2** | **74.6** | **72.8** | **79.1** | **76.4** | **68.8** | **69.5** | **69.1** | **63.1** | **63.6** | **63.4** |

**Table 2: Statistical results of different datasets. NS and MT denote Negative Scenes and Multiple Targets, respectively.**

| Dataset | Single Scene | Multiple Scenes | Grouped | NS | MT |
|---|---|---|---|---|---|
| Sr3D [2] | 83572 | - | ✗ | ✗ | ✗ |
| G-Sr3D-ST | 21784 | 43416 | ✓ | ✓ | ✗ |
| G-Sr3D-MT | 16758 | 33518 | ✓ | ✓ | ✓ |

scenes; b) ST w/ negatives; c) multiple targets (MT) w/o negatives; d) MT w/ negatives. Note that b) and d) correspond to the original G-Sr3D-ST/MT datasets where negative scenes can exist in a group, while a) and c) are simpler cases where we remove the negative scenes in these datasets. For each case, we divide the dataset into "single" and "multiple" subsets, depending on whether there are multiple 3D scenes in a group.

**Implementation Details.** We set the number of transformer layers to 4 for scene-text encoding and multi-modal fusion, and 2 for LCAM and CFRM. For all modules, the hidden size $D$ is set to 768 and the number of attention heads is set to 12. The number of points $N$ in object proposals is set to 1024. The threshold $\tau_s$ and $\tau_o$ are set to 0.2 and 0.1, respectively. The reduction ratio $r$ is set to 16. For curriculum learning, the default pacing function is linear function, and $\kappa = 8$. Following [63], we use the AdamW [35] optimizer to optimize the model with the initial learning rate of $1e^{-4}$. The model is trained for 80 epochs with batch size of 8.

## 4.3 Comparison with SOTA Methods

**Results on the group-wise setting.** We first compare GNL3D with existing 3D object grounding models on the re-built G-Sr3D-ST/MT datasets in different cases (see Tab. 1). We focus on four two-stage methods that perform well on Sr3D. These methods are adapted to our task and applied to each scene in a group individually. To better understand our GNL3D's performance, we further extend the SOTA method 3D-VisTA [63] to the group-wise setting as a competitive baseline. Specifically, inspired by CoADNet [58] (a 2D group-wise learning method), we insert Global Self-Attention [43] (GSA) layers between the scene encoding and multi-modal fusion modules in 3D-VisTA, which take object features of all scenes in a group as input to learn inter-scene visual correspondences for group-wise relationship modeling. The results reveal some interesting points. 1) Our GNL3D significantly outperforms existing single-scene 3D grounding methods in all cases, due to the proposed LCAD mechanism, which explicitly exploits the intra-group visual

**Table 3: Comparison of different methods on Sr3D val set.**

| Method | Easy | Hard | View-Dep | View-Indep | **All** |
|---|---|---|---|---|---|
| 3DVG-Trans [61] | 54.2 | 44.9 | 44.6 | 51.7 | 51.4 |
| LanguageRefer [41] | 58.9 | 49.3 | 49.2 | 56.3 | 56.0 |
| TransRefer3D [18] | 60.5 | 50.2 | 49.9 | 57.7 | 57.4 |
| SAT [50] | 61.2 | 50.0 | 49.2 | 58.3 | 57.9 |
| LAR [3] | 63.0 | 51.2 | 50.0 | 59.1 | 59.4 |
| 3D-SPS [36] | 56.2 | 65.4 | 49.2 | 63.2 | 62.6 |
| MVT [21] | 66.9 | 58.8 | 58.4 | 64.7 | 64.5 |
| BUTD-DETR [23] | - | - | 53.0 | - | 67.0 |
| ViL3DRel [10] | 74.9 | 67.9 | **63.8** | 73.2 | 72.8 |
| NS3D [19] | - | - | 62.0 | - | 62.7 |
| ViewRefer [17] | 69.7 | 61.7 | 56.9 | 67.8 | 67.2 |
| EDA [46] | 70.3 | 62.9 | 54.1 | 68.7 | 68.1 |
| 3D-VisTA †[63] | 72.1 | 63.6 | 57.9 | 70.1 | 69.6 |
| GNL3D (Sr3D) | 72.8 | 64.0 | 58.0 | 70.6 | 70.1 |
| GNL3D (+G-Sr3D-ST) | **75.4** | **68.0** | 58.2 | **73.8** | **73.2** |
| Δ | +2.6 | +4.0 | +0.2 | +3.2 | +3.1 |

connections. 2) 3D-VisTA + GSA achieves clearly inferior results than GNL3D and is even worse than its base model, which suggests that without language as a guide, it is hard to effectively exploit the intra-group visual information, further verifying the superiority of our method. 3) GNL3D achieves SOTA results on both "single" and "multiple" subsets, indicating that our LCAD mechanism can boost performance not only in the multi-scene case but also in the conventional single-scene case.

**Results on the traditional setting.** To further study the effectiveness of our method on the traditional 3D object grounding task, we compare our GNL3D with recent models on Sr3D (see Tab. 3). † denotes that 3D-VisTA is trained from scratch without pre-training on additional data. We report results of GNL3D trained on Sr3D from scratch or pre-trained on the G-Sr3D-ST dataset without negative scenes. The main observations are as follows: 1) GNL3D (Sr3D) only achieves slightly better results than the base model 3D-VisTA. 2) Pre-training on the reorganized G-Sr3D-ST dataset significantly improves the performance of GNL3D, which sets a new record on the Sr3D benchmark without additional training data. The performance gap suggests that with group-wise training, the proposed LCAM learns to extract effective visual features even from a single 3D scene to form a precise target concept, and the CFRM can exploit these homo-modal target features to enhance the feature discriminability and bridge the modality gap.

**Table 4: Ablation studies of main designs in GNL3D.**

|     | LCAM | CFRM | ST w/o Negatives | | |
|-----|------|------|--------|----------|-----|
|     |      |      | Single | Multiple | **All** |
| (a) | None | None | 65.7 | 72.6 | 69.7 |
| (b) | w/o SCA | Full | 68.8 | 77.0 | 73.5 |
| (c) | w/o CCA | Full | 68.1 | 76.4 | 72.8 |
| (d) | Full | w/o CAWG | 67.7 | 75.6 | 72.2 |
| (e) | Full | w/o LSAP | 69.3 | 77.4 | 73.9 |
| (f) | Full | w/o SSCA | 69.1 | 77.3 | 73.8 |
| (g) | Full | Full | **69.9** | **78.2** | **74.6** |

**Table 5: Ablation studies of the CL strategy in LCAM.**

|     | Affected Branch | Pacing Function | ST w/ Negatives | | |
|-----|-----------------|-----------------|--------|----------|-----|
|     |                 |                 | Single | Multiple | **All** |
| (a) | None | None | 69.8 | 75.8 | 73.2 |
| (b) | SCA+CCA | Linear | 71.1 | 77.2 | 74.6 |
| (c) | CCA | Logarithmic | 71.7 | 78.0 | 75.3 |
| (d) | CCA | Exponential | 72.4 | 78.6 | 75.9 |
| (e) | CCA | Linear | **72.8** | **79.1** | **76.4** |

## 4.4 Ablation Studies

To investigate the effectiveness of our main designs in GNL3D, we conduct ablation studies on G-Sr3D-ST without negative scenes. Concretely, the LCAM includes Semantic-level Consensus Aggregation (SCA) and Contextual Consensus Aggregation (CCA). CFRM includes Consensus-Adaptive Weight Generation (CAWG), Layer-Specific Attentive Pooling (LSAP), and Scene-Specific Consensus Adaptation (SSCA). We selectively discard these designs to construct ablation models and report the results in Tab. 4. From these results, we can find that the full model outperforms all ablation models, validating each component is helpful for group-wise 3D object grounding. The baseline (a) without LCAD degrades to the base model 3D-VisTA and has the same results. (b) and (c) show that removing CCA causes worse performance degradation, demonstrating capturing contextual visual features is essential for consensus aggregation in 3D group-wise learning. In case (d), we remove CAWG and conduct consensus distribution based on objects-consensus cross-attention, which reduces the performance greatly, showing the effectiveness of our dynamic attention-based CFRM.

To demonstrate the effectiveness of the proposed curriculum learning strategy, we further conduct detailed ablation studies on G-Sr3D-ST with negative scenes. Specifically, we compare GNL3D models trained with the CL strategy applied on different LCAM branches using different pacing functions. As shown in Tab. 5, (a) denotes the model trained without any CL strategy, which performs significantly worse than other models, verifying the CL strategy can consistently boost performance in the presence of negative scenes. Comparing (b) to (e), we find that applying CL on SCA leads to severely decreased performance, which suggests that although negative scenes cannot provide direct contextual visual clues about the described object, they may contain useful semantic-level information. Cases (c)-(e) show that models trained with linear and exponential pacing functions slightly outperform the model trained with logarithmic function, indicating that the model should focus

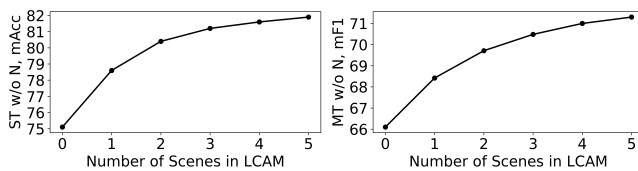

**Figure 5: Effect of the number of 3D scenes fed into LCAM.**

**Query:** Looking at the front of the bed, select the nightstand that is to the left of it.

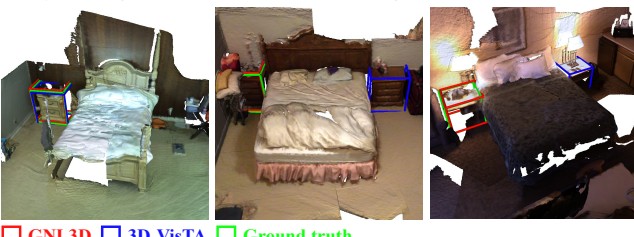

□ **GNL3D**    □ **3D-VisTA**    □ **Ground truth**

**Figure 6: Group-wise 3D object grounding example results.**

more on the positive scenes in the beginning of training to learn valid consensus patterns.

Besides, we explore the effect of the number of 3D scenes fed into LCAM in Fig. 5. We evaluate GNL3D on a subset of G-Sr3D-ST/MT without negatives that only contains groups with more than 5 scenes, and we sample different numbers of scenes in a group as the input of LCAM. The number 0 denotes the base model without LCAD. As can be seen, with group-wise training, GNL3D with even a single randomly sampled 3D scene fed into LCAM can outperform the base model. When increasing the number of scenes that LCAM can utilize, our model performances get better.

## 4.5 Qualitative Results

We qualitatively compare our GNL3D to its base model 3D-VisTA on G-Sr3D-ST without negatives and display a typical example in Fig. 6. By capturing the additional intra-group vision-vision connections via the proposed LCAD mechanism, GNL3D can better understand the described object and make more accurate predictions.

## 5 CONCLUSION

In this work, we present a more realistic group-wise setting for the 3D object grounding task, which extends the traditional setting to a group of related 3D scenes, allowing a flexible number of target objects to exist in each scene. We propose a baseline method named GNL3D to tackle this new task, which extends the conventional 3D object grounding pipeline with a novel language-guided consensus aggregation and distribution mechanism to explicitly capture the language-vision and intra-group vision-vision connections for better understanding of the described 3D object. To validate the effectiveness of the proposed method, we introduce the G-Sr3D-ST/MT datasets by reorganizing and enhancing the ReferIt3D benchmark. Extensive experiments demonstrate that our method achieves state-of-the-art results on both the group-wise setting and the traditional 3D object grounding task.

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
