# OpenReview forum: "Advancing 3D Object Grounding Beyond a Single 3D Scene"
_acmmm.org/ACMMM/2024/Conference — MM2024 Poster_

### Official Review · Reviewer_gqSX · 2024-05-22

**Rating:** 3
**Confidence:** 2

**Summary:**

This paper proposes to extend the task of multiple 3D object grounding, i.e., the task of localizing instances of a 3D object given a textual description, to groups of scenes instead of a single scene. The main motivation behind the proposed task is that similar scenes, although not exactly equal (e.g., bedrooms), have correlations, which can help object localization. After proposing the task, a baseline method named GNL3D (Grouped Neural Listener for 3D grounding) is presented. The model consists of an end-to-end neural architecture based on the cross-attention mechanism to perform the multimodal fusion between text and point cloud representations. Additionally, a curriculum strategy is proposed to help the multimodal fusion proceed more smoothly in cases where the scene group contains scenes that do not have the target object inside. Various experiments are performed to validate the proposed model, while ablation studies are used to validate the different design decisions.

**Strengths:**

- The paper is quite clear and the provided explanations render the content self-contained.

- The model design follows intuitive concepts, which later seem to translate well into experimental results.

- Compared to state-of-the-art approaches, the experimental results are great in the tested settings.

- The masking strategy for negative scenes is a clever trick with which to apply a CL strategy, and indeed it seems to give a performance boost from the ablation studies.

**Limitations:**

- The main limitation I see in this paper is that it is difficult to understand why such a scenario is needed and what it brings to the table compared to multiple objects grounded in a single scene. While the extension from localizing a single object to multiple ones is intuitive and straightforward, going to groups of scenes does not seem to be the same. Given that the task is solved via an NN architecture, information about the localization of objects in similar rooms is implicitly encoded in the learned representations of the network. To clarify things with an example, consider how, for any task, such as image classification, the classifier learns similar representations for images of a similar class. I do not see how training with a multi-target objective, but a single-scene scenario, would be that different from what is proposed in the current method. This whole discussion is motivated by the comparison with the traditional 3D grounding approach in SR3D but not with Multi3DRefer [1], a dataset and task upon which they also have an elongated discussion in lines 128-144. Such a comparison is, in my opinion, essential, as it could at least provide an empirical reason for why the group setting is as critical as the authors claim.

- The proposed approach is presented as a baseline, but it is indeed a complex architecture comprised mainly of cross-attention operations. This raises questions about the model's efficiency and scalability and this task in general. Therefore, a note or discussion on this aspect compared to other models would benefit the presentation of the method.

- As a final but minor point, why is the masking strategy only used within the contextual aggregation? Having a semantic level aggregation in scenes where the object does not exist from the beginning, when this could be avoided, seems like a strange design choice.

[1] Yiming Zhang, ZeMing Gong, and Angel X Chang. 2023. Multi3drefer: Grounding text description to multiple 3d objects. In Proceedings of the IEEE/CVF International Conference on Computer Vision. 15225–15236.

**Suitability:**

3

---

### Official Review · Reviewer_DjyL · 2024-05-24

**Rating:** 4
**Confidence:** 3

**Summary:**

The objective is to perform 3D Object Grounding in a group-wise setting and ground a flexible number of object in each 3D scene of the group, guided by a language query.  Instead of localizing 3d objects in each scene individually, the proposed method target to create a comprehensive representation of target object guided from multiple 3D scenes.

**Strengths:**

This work utilizes language guidance to aggregate information across different 3D scenes for better 3D visual grounding. The motivation to extract and aggregate visual features guided from a common referring sentence is appreciated. The authors have used repeated “target-relation- anchor” occurrences across multiple 3D scenes in Sr3D  to  re-package the dataset having groups of positive scenes. The work has reported best objective numbers on Sr3D val dataset.

**Limitations:**

The work in [59] also can handle multiple objects or no objects in a scene. The authors have mentioned the limitations of applying [59] individually to each scene in the introduction. However [59] is ignored in the experimental section. The objective comparison with [59] is required to bring about the advantage  of the proposed method in group-wise setting.

The work uses repeated “target-relation- anchor” occurrences across multiple 3D scenes to define a group of positive scenes. The objective is to extract robust features utilizing commonality between  various 3D scenes in the context of a common grounding target. Grounding an object based on only appearance do not need the complexity of a group of 3D scenes. For example, in Fig 1 with query “Find the monitor that is on top of a desk.”  the grounding can be done by learning the appearance of monitor only. Learning of spatial layout of  monitor is not required as there is no ‘monitor on floor’ kind of  scene. The choice of a hard negative is not explored in the work.

The information from multiple scenes are simply concatenated and fed to attention mechanism to learn cross-corelation among different scenes.  For various given “target-relation- anchor” the number of positive scenes will vary . It is not clear how the same curriculum learning strategy will work for the varying number of positive scenes across different referring sentences.

During inference time if the grounding expression do not create many positive scenes, the proposed method will not have a special advantage.

[59] Multi3drefer: Grounding text description to multiple 3d objects.

**Suitability:**

3

---

### Official Review · Reviewer_UmrW · 2024-05-24

**Rating:** 3
**Confidence:** 3

**Summary:**

The paper presents a novel framework called GNL3D for the task of Group-wise 3D Object Grounding, which aims to localize objects described by natural language across multiple 3D scenes. This approach addresses the limitation of traditional 3D object grounding tasks that assume the presence of the object in a single scene. GNL3D uses a Language-guided Consensus Aggregation Module (LCAM) to aggregate visual features across scenes, creating a consensus representation that improves object localization accuracy.

**Strengths:**

1. The LCAM effectively utilizes intra-group visual information and language descriptions to form a comprehensive visual consensus, enhancing the model's ability to locate objects accurately across varied scenes.
2. By reorganizing the ReferIt3D dataset and proposing new evaluation metrics, the study provides a solid foundation for benchmarking the performance of group-wise 3D object grounding methods.

**Limitations:**

1. Insufficient Motivation: The choice between focusing on multiple objects in one scene or across multiple scenes should be driven by specific user requirements. The paper asserts that the group-wise setting is inherently more rational, which is debatable. The task of distinguishing between similar objects within a single scene based on complex language descriptions can be even more challenging and arguably requires a higher degree of language understanding and spatial reasoning.
2. Unreasonable Design Assumptions: The approach assumes the availability of several related scenes during inference, which may not always be feasible. This requirement for multiple similar scenes to aid the inference process can be a significant limitation if the model needs to be applied to a new room or an environment without prior similar contextual data.

**Suitability:**

2

---

### Meta-Review · Area_Chair_FzWa · 2024-07-04

**Recommendation:** Accept (Poster)
**Confidence:** 5

**Metareview:**

This paper introduces a novel framework named GNL3D for Group-wise 3D Object Grounding. This method aims to localize objects described by natural language across multiple 3D scenes, addressing the limitations of traditional 3D object grounding tasks that typically focus on single-scene localization. The framework employs a Language-guided Consensus Aggregation Module (LCAM) to aggregate visual features across scenes, forming a consensus representation to enhance object localization accuracy. The reviewers acknowledge the novelty and potential impact of the proposed framework, particularly its innovative approach to aggregating visual features across multiple scenes. However, they also highlight several critical limitations, such as the need for stronger motivation, assumptions on scene availability, and the lack of direct comparisons with similar methods.